# Adsorptive Cathodic Stripping Voltammetry for Quantification of Alprazolam

**DOI:** 10.3390/molecules26102958

**Published:** 2021-05-16

**Authors:** Waree Boonmee, Kritsada Samoson, Janjira Yodrak, Adul Thiagchanya, Apichai Phonchai, Warakorn Limbut

**Affiliations:** 1Division of Health and Applied Sciences, Faculty of Science, Prince of Songkla University, Hat Yai, Songkhla 90112, Thailand; Waree_b35@hotmail.com (W.B.); 5910220008@email.psu.ac.th (K.S.); adul.t@psu.ac.th (A.T.); papichai_13@hotmail.com (A.P.); 2Forensic Innovation Center, Prince of Songkla University, Hat Yai, Songkhla 90112, Thailand; 3Center of Excellence for Trace Analysis and Biosensor, Prince of Songkla University, Hat Yai, Songkhla 90112, Thailand; 4Center of Excellence for Innovation in Chemistry, Faculty of Science, Prince of Songkla University, HatYai, Songkhla 90112, Thailand; 5Satun Provincial Police Forensic Science, Police Forensic Science Center 9, Office of Police Forensic Science, Royal Thai Police, Mueangsatun, Satun 91000, Thailand; Yodrak.J@gmail.com

**Keywords:** electrochemical pretreatment, sensor, alprazolam, adsorptive cathodic stripping voltammetry

## Abstract

A simple and highly sensitive electrochemical sensor was developed for adsorptive cathodic stripping voltammetry of alprazolam. Based on an electrochemically pretreated glassy carbon electrode, the sensor demonstrated good adsorption and electrochemical reduction of alprazolam. The morphology of the glassy carbon electrode and the electrochemically pretreated glassy carbon electrode were characterized by scanning electron microscopy/energy dispersive X-ray spectroscopy, Fourier transform infrared spectroscopy, cyclic voltammetry, and electrochemical impedance spectroscopy. The electrochemical behaviors of alprazolam were determined by cyclic voltammetry, and the analytical measurements were studied by adsorptive cathodic stripping voltammetry. Optimized operational conditions included the concentration and deposition time of sulfuric acid in the electrochemical pretreatment, preconcentration potential, and preconcentration time. Under optimal conditions, the developed alprazolam sensor displayed a quantification limit of 0.1 mg L^−1^, a detection limit of 0.03 mg L^−1^, a sensitivity of 67 µA mg^−1^ L cm^−2^ and two linear ranges: 0.1 to 4 and 4 to 20 mg L^−1^. Sensor selectivity was excellent, and repeatability (%RSD < 4.24%) and recovery (82.0 ± 0.2 to 109.0 ± 0.3%) were good. The results of determining alprazolam in beverages with the developed system were in good agreement with results from the gas chromatography–mass spectrometric method.

## 1. Introduction

Alprazolam is a psychotropic drug in the benzodiazepine class [1,2]. It produces muscle-relaxant, hypnotic, anticonvulsant, and sedative [3] effects. It is widely used to treat alcohol withdrawal syndromes, insomnia, psychiatric disorders, depression, panic attacks, and generalized anxiety disorders [4,5]. Alprazolam has been increasingly involved in cases of murder and sexual abuse due to its widespread availability and toxic effects [2,4]. Alprazolam overdose causes muscle numbness, drowsiness, fainting, coordination problems, and death [6]. Therefore, the development of a simple and highly sensitive method of determining alprazolam is of great interest to forensic toxicologists and clinicians.

Developed analytical methods for the detection of alprazolam include high performance liquid chromatography [7], gas chromatography [8], UV-vis spectroscopy [9], and electrochemical techniques [10,11,12]. Electrochemical techniques are simple, inexpensive, and provide fast response times and good sensitivity [13,14,15,16,17]. Adsorptive stripping voltammetry (AdSV) is an electrochemical technique that is widely used for the analysis of trace organic compounds and various elements [18,19,20]. Since the target analyte accumulates on the surface of a working electrode in a preconcentration step, low limits of detection and high sensitivity are possible with AdSV [21,22,23]. Electrochemical analysis frequently involves glassy carbon electrodes (GCE) due to their low background current, wide potential window, easy surface modification, hardness, and low cost [24,25,26,27]. However, the use of bare GCEs is likely to produce sluggish electron transfer [14,26,27]. Furthermore, the limited number of active sites on the surface of GCEs can result in poor selectivity and sensitivity toward target analytes [24].

Improving the performance of GCE has been attempted through various pretreatment techniques [28]. These techniques have included in situ laser irradiation [29], radio frequency plasma pretreatment [30], and electrochemical pretreatment [31,32]. In particular, electrochemical pretreatments are receiving increasing attention because they can be rapid, environment-friendly, simple, and inexpensive. In addition, electrochemical pretreatment of GCEs roughens the polished surface of the electrode and populates it with the oxygen-containing functional groups of graphite oxide [31,32,33]. These functional groups can interact easily with alprazolam via hydrogen bonding, both of which can increase the adsorption of alprazolam on the electrode surface. Increasing the adsorption of an analyte on the electrode improves the sensitivity and detection limit of a sensing system.

To the best of our knowledge, the present work is the first report of the determination of alprazolam using an EPGCE. The EPGCE was fabricated for a simple and highly sensitive electrochemical sensor based on adsorptive cathodic stripping voltammetry (AdCSV). The EPGCE was used as an adsorbent and working electrode to determine the drug alprazolam by AdCSV. The EPGCE was prepared by a simple electrochemical method. The morphology of the GCE and the EPGCE were characterized by scanning electron microscopy/energy dispersive X-ray spectroscopy (SEM/EDX), Fourier transform infrared spectroscopy (FT-IR), cyclic voltammetry (CV), and electrochemical impedance spectroscopy (EIS). The electrochemical behaviors of alprazolam were determined by CV, and the analytical measurements were obtained by AdCSV. Operational parameters of the EPGCE were optimized for the determination of alprazolam. These included the electrochemical pretreatment, preconcentration potential, and preconcentration time. The analytical performances of the alprazolam sensor were also evaluated, including linear range, limit of detection, limit of quantification, repeatability, selectivity, and analysis of alprazolam in beverage samples.

## 2. Results and Discussion

### 2.1. Chemicals for the Electrochemical Pretreatment

The chemical media used for the electrochemical pretreatment of the GCE are an important factor in the adsorption and electrochemical reduction of alprazolam, since they are involved in the production of oxygen-containing functional groups and enhancement of the electroactive surface area of the electrode surface. The effects of sulfuric acid, perchloric acid, and sodium hydroxide were investigated on the detection of alprazolam at 4 mg L^−1^ (Figure 1). Pretreatment with sulfuric acid produced the electrode with the highest current response for alprazolam determination. Thus, sulfuric acid was employed for the electrochemical pretreatment of the GCE to detect alprazolam molecule.

### 2.2. Characterization

SEM was used to identify the surface morphology of the GCE and the EPGCE (GCE pretreated with sulfuric acid). As shown in Figure 2A, the SEM image of the surface morphology of the GCE before pretreatment presents clearly smooth. After the GCE was electrochemically pretreated with sulfuric acid (Figure 2B), its surface displays higher roughness, looking like the hierarchal surface structure compared to the GCE before pretreatment.

EDX was further used to identify elements on the surface of the GCE (Figure 2C) and EPGCE (Figure 2D). Since the EPGCE produced a significantly stronger oxygen signal in the EDX spectrum than the GCE, the presence of oxygen-containing functional groups on the surface of the EPGCE was confirmed [34].

Furthermore, the oxygen-containing functional groups information on the GCE surface before and after electrochemical pretreatment in sulfuric acid was characterized by the FT-IR technique, as shown in Figure 2E. Before GCE electrochemical pretreatment (Figure 2E-a), small peaks at 2921, 2860, and 1621 cm^−1^, which attribute to stretching band of -C-H and C=C in the structure of carbon material were observed [18,35], indicating a less functional group on the pristine GCE surface. In contrast, after GCE was electrochemically pretreated in sulfuric acid at 1.8 V (Figure 2E-b), stretching bands were more pronounced than pristine GCE, which can be observed in the bands of the oxygen-containing functional groups in the region of 1800 to 700 cm^−1^, which can be assigned to -C=O, -C=C, -COO^−^, -C-O-C, and C-C stretching bands and -OH bending modes in ether, lactones, carboxyl, and phenolic structures [15,18,33]. The strong intensities of those bands after electrochemical pretreatment of GCE in sulfuric acid confirmed that the oxygen-containing functional groups were successfully functionalized on the GCE surface during electrochemical pretreatment [33].

### 2.3. Electrochemical Behavior

The electrochemical property of the GCE and the EPGCE was evaluated by EIS and CV technique, which was performed in 0.10 M KCl containing 5.0 mM Fe(CN_6_)^3−/4−^. Figure 3A displays Nyquist plots of the GCE (trace a) and the EPGCE (trace b). It can be seen that the charge transfer resistances that were determined from the Nyquist plots of the GCE (trace a) and the EPGCE (trace b) were 2.72 and 0.44 kΩ, respectively, indicating that the electrochemical pretreatment can improve surface active area led to enhance the conductivity and electron transfer of the GCE.

Figure 3B shows the CV response of the GCE and the EPGCE. For the unpretreated GCE (trace a), well-defined redox peaks for the Fe(CN_6_)^3−/4−^ redox marker are observed, with peak-to-peak (ΔE_p_) is 237 mV. For the EPGCE (trace b), the redox peak current is increased along to decreasing of peak-to-peak separation (ΔE_p_) (128 mV) due to its high surface area of the EPGCE as a result of the electrochemical pretreatment, which will enhance electron transfer and conductivity at the electrode surface.

The electrochemical behavior of alprazolam at the GCE and the EPGCE were studied in BR buffer at pH 9 by CV at scan rate 100 mV s^−1^ with potential scanning from −0.80 to −1.20 V to cover the reduction peak of alprazolam and to avoid the interference from reduction reaction of the oxygen-containing functional group on the pretreated GCE. In the case of without alprazolam, the background current of the EPGCE (Figure 3C (trace b)) was significantly greater than the background current of the GCE (Figure 3C (trace a)). This was likely due to the larger surface area with high oxygen-containing functional groups provided by the rough surface created during electrochemical pretreatment [33,36]. After adding 4 mg L^−1^ alprazolam into the electrolyte, no significant change occurred in the current peak of the GCE (Figure 3D (trace a)). However, at the EPGCE, the addition of alprazolam produced a strong cathodic peak response at a potential of −1.06 V (Figure 3D (trace b)). This was attributed to increased adsorption of alprazolam on the larger surface area of the EPGCE. In addition, the EPGCE possessed the oxygen-containing functional groups of graphite oxide, which could enhance the adsorption of alprazolam on the electrode surface via hydrogen bonding. The electrochemical reduction mechanism of alprazolam has been previously described [11,37]. The azomethine functional group of alprazolam is reduced with two protons and two electrons at the double bond position (Figure 3D (inset)).

### 2.4. Effect of Scan Rate

The electrochemical behavior of a solution of 4 mg mL^−1^ of alprazolam in BR buffer was studied by CV at various scan rates from 50 to 400 mV s^−1^ (Figure 3E). The cathodic peak current for the reduction of alprazolam was correlated linearly with the scan rate (v). The linear regression equation was I_pc_ = (82.0 ± 1.0) × −(2.0 ± 0.4) and the correlation coefficient was 0.9982 (Figure 3F). This result confirmed that the electrochemical reduction of alprazolam at the EPGCE was controlled by an adsorption process.

### 2.5. Optimization

To obtain good adsorption and electrochemical reduction performances of alprazolam at the EPGCE, the operational conditions were optimized. Each parameter was optimized with 0.25, 0.5, 1, 2, and 4 mg L^−1^ alprazolam in BR buffer. The electrochemical pretreatment parameters were the concentration of the sulfuric acid medium and the duration of pretreatment; the preconcentration parameters were potential and time. The conditions that produced the highest sensitivity were considered optimal.

#### 2.5.1. Electrochemical Pretreatment

Concentration of Sulfuric Acid

The effect of sulfuric acid concentration on the electrochemical pretreatment of the GCE was studied from 50 to 150 mmol L^−1^ (Figure 4A). The sensitivity of the response reached a maximum at the electrode pretreated in 100 mmol L^−1^ sulfuric acid. Therefore, 100 mmol L^−1^ sulfuric acid was employed for further pretreatment procedures.

Pretreatment Time

The influence of the duration of electrochemical pretreatment was evaluated from 5.00 to 17.50 min (Figure 4B). Sensitivity increased with pretreatment time from 5.00 to 12.50 min and then gradually decreased. The decrease in sensitivity might have been due to an excessive amount of the oxygen-containing functional groups of graphite oxide on the surface of the EPGCE. Too much graphite oxide could reduce electrical conductivity and the efficiency of electron transfer [35,38,39]. Therefore, a duration of 12.50 min was used for subsequent electrode pretreatments.

#### 2.5.2. Preconcentration Potential and Preconcentration Time

The preconcentration step in the AdCSV procedure was extremely important since a suitable preconcentration potential and preconcentration time could enhance the adsorption of alprazolam on the surface of the EPGCE, which would improve the detection limit and sensitivity. The influence of the preconcentration potential applied to the EPGCE was evaluated from 0.00 to 0.20 V (Figure 4C). The results indicated that the highest sensitivity of response occurred at 0.10 V. The effect of preconcentration time was then evaluated between 120 and 960 s using a potential of 0.10 V (Figure 4D). The sensitivity of response increased as time increased from 120 to 480 s. Beyond 480 s, sensitivity gradually decreased. Thus, a potential of 0.10 V and a duration of 480 s were selected as the optimal preconcentration conditions.

### 2.6. Analytical Performance

#### 2.6.1. Linearity, Limit of Detection, and Limit of Quantification

The analytical performances of the EPGCE for alprazolam measurement were evaluated under the optimal conditions of AdCSV measurement by using the same EPGCE detect each concentration of alprazolam between 0.1 and 20 mg L^−1^. The cathodic peak currents produced at these alprazolam concentrations are presented in Figure 5A. The calibration plot of cathodic peak current versus concentration of alprazolam produced two linear ranges: one from 0.1 to 4 mg L^−1^ and another from 4 to 20 mg L^−1^ (Figure 5B). The difference in sensitivity was due to the different ways alprazolam was adsorbed on the surface of the EPGCE. The alprazolam could be adsorbed as a monolayer on the EPGCE surface, in which case sensitivity was higher over the linear range produced. Alprazolam at higher concentrations could also be adsorbed as a multilayer, in which case sensitivity was lower over the linear range produced [14,40,41]. The detection limit (LOD) and quantification limit (LOQ) were calculated by using 3.3σ/S for LOD and 10σ/S for LOQ (σ is the standard deviation of blank and S is the slope of the calibration curve) and were found to be 0.03 and 0.1 mg L^−1^, respectively. The analytical performances of the developed EPGCE system were compared with performances of other techniques for the determination of alprazolam (Table 1). The developed alprazolam sensor showed a wide linear range and the lowest limit of detection. These good analytical performances could be attributed to the electrochemical pretreatment of the developed sensor, which produced oxygen-containing functional groups of graphite oxide on the surface of the electrode. These functional groups enhanced the adsorption of alprazolam in the preconcentration step of AdCSV, which led to good performance in the determination of alprazolam.

#### 2.6.2. Repeatability

The repeatability of the EPGCE fabrication was evaluated by measuring the cathodic peak currents of standard alprazolam solutions at 0.25, 0.5, 1, 2, and 4 mgL^−1^ at the same EPGCE (Figure 5C). Six measurements were taken at each concentration (n = 3) for a total of 18 measurements. The same concentrations were also determined once using six different fabricated EPGCEs (Figure 5D). From the measurements taken using the same electrode, the relative standard deviations (RSDs) of response across all concentrations varied from 1.35% to 3.74% (Figure 5C). The relative standard deviations of measurements using the six different EPGCEs varied from 1.31% to 4.24% (Figure 5D). The levels of repeatability obtained from the single EPGCE and the six different EPGCEs were acceptable according to the AOAC guideline of 7.3% for 10 mg L^−1^ [40]. These results showed the good repeatability of the developed alprazolam sensor.

#### 2.6.3. Selectivity

The interference species in beverage samples can include ascorbic acid, citric acid, sucrose, and paracetamol. The influences of these possible interference species were studied by mixing each species with 4 mg L^−1^ of alprazolam in an electrochemical cell for determination under optimal conditions. The tolerance limit of each interfering compound was defined as the highest interfering concentration that caused an error of less than ±5% in the alprazolam measurement. No interference was detected on the determination of alprazolam in the presence of these compounds, including 50-fold ascorbic acid and citric acid, 40-fold paracetamol, and 30-fold sucrose concentrations (Figure 5E). In addition, selectivity of the EPGCE was also tested based on sensitivity under the optimal conditions by detecting the individual standard alprazolam, phenazepam, clonazepam, and diazepam from the benzodiazepine class and tramadol at the same series concentration of 0.25, 0.5, 1, 2, and 4 mgL^−1^ (Figure 5F). The results showed that the EPGCE could detect other benzodiazepines with different sensitivity. The sensitivity for alprazolam detection provides higher than the sensitivity for phenazepam, clonazepam, and diazepam at 3.9-fold, 12.7-fold, and 24.8-fold, respectively. This result indicated that the EPGCE can not only detect alprazolam, but also can detect other benzodiazepines. Nonetheless, in cases of murder and sexual abuse, the alprazolam drug was individually used. Moreover, the EPGCE can also be used to screen the alprazolam if presence other benzodiazepines via identification of the reduction peak; the reduction peaks of alprazolam appeared at −1.100 V, but the reduction peaks of phenazepam, clonazepam, and diazepam appeared at −1.240, −1.205, and −1.205 V, respectively. In the case of tramadol, the EPGCE cannot detect due to an oxidation reaction occurring at the positive potential [41,44,45], and therefore tramadol produced no signal for detection. These results indicated that the developed sensor could be applied for detecting individual alprazolam in beverage samples and screening alprazolam if other benzodiazepines were present via identification of its reduction peak.

#### 2.6.4. Real Sample Analysis

To evaluate the performance of the developed sensor for practical application, alprazolam in six beverage samples was determined: Pepsi Max Taste, Smirnoff Black Ice, Eristoff vodka, Coke Lite, Orange Big, and Full Moon Wine, which were from a local supermarket using standard addition method. The recoveries were evaluated by spiking each beverage sample with standard alprazolam solution. The electrochemical cell of the sensor was filled with 5.0 mL of BR buffer, and final sample concentrations were 4, 8, and 16 mg mL^−1^. The recoveries obtained from alprazolam determination ranged from 82.0 ± 0.2 to 109.0 ± 0.3%, which were within the AOAC guidelines (Table 2). The standard addition GC–MS method was used to detect alprazolam in three beverage samples (Pepsi Max Taste, Eristoff vodka, and Smirnoff Black Ice). There were no significant differences at a 95% confidence level (*p* > 0.05) between alprazolam concentrations detected by the proposed method and by GC–MS method (Table 3).

## 3. Materials and Methods

### 3.1. Materials

Alprazolam, ascorbic acid, boric acid, acetic acid, and sodium hydroxide were from Ajax Finechem (Taren Point, Australia). Phosphoric acid and sulfuric acid were from J.T. Baker (Pennsylvania, USA). Perchloric acid was from Sigma-Aldrich (St. Louis, MO, USA) All solutions were prepared using deionized water (18.2 MΩ cm^−1^) (Barnstead^TM^ Easy Pure^TM^ II water purification system, Thermo Scientific^TM^, Waltham, USA).

### 3.2. Apparatus

All electrochemical analyses were conducted using an Autolab 910 PSTAT Mini controlled by the PSTAT software (Metrohm Autolab B.V., Utrecht, The Netherlands). The three-electrode system used the electrochemically pretreated GCE (EPGCE) as the working electrode, an Ag/AgCl reference electrode, and a platinum wire (Ø = 1 mm) auxiliary electrode, and all these electrodes were from Metrohm, Netherlands. The morphology of the GCE and the EPGCE were evaluated by SEM-EDX (Quanta 400, FEI, Hillsboro, Oregon, USA, USA).

### 3.3. Electrode Preparation

A glassy carbon electrode (GCE, Ø = 3 mm, Metrohm, Switzerland) was polished to a smooth mirror finish with a sequence of 1.5, 0.5, and 0.05 μm alumina slurries. The GCE was then rinsed and sonicated with double deionized water for 5 min and dried with a pure nitrogen gas stream. The polished GCE was pretreated by electro-oxidation in sulfuric acid at 1.8 V for 750 s (Figure 6). To obtain an electrode that provided good adsorption and electrochemical responses from the target analyte, pretreatment media, medium concentration, and duration of electrochemical pretreatment were evaluated.

### 3.4. Electrochemical Measurements

The electrochemical behavior of alprazolam was studied by CV from −1.20 to −0.80 V at a rate of 100 mV s^−1^ in an electrochemical cell filled with 5.0 mL of Britton Robinson (BR) buffer at pH 9. To begin AdCSV analysis, a few microliters of alprazolam standard solution were injected into the electrolyte and the current response of alprazolam reduction was measured. AdCSV comprises two steps: preconcentration and stripping. In the preconcentration step, alprazolam was allowed to accumulate on the surface of the electrode for set time intervals at set potentials while being magnetically stirred. Therefore, the important parameters AdCSV analysis included preconcentration potential and preconcentration time were evaluated for optimization. In the stripping step, stirring was halted for 10 s to enable chemical equilibrium as the alprazolam accumulated on the surface of the electrode was reabsorbed into solution before voltametric scanning from −0.80 to −1.30 V at a scan rate of 100 mV s^−1^. The current response due to the reduction of alprazolam was recorded.

### 3.5. Real Sample Analysis

Six samples of alcoholic and non-alcoholic beverages were obtained from a local supermarket. The standard alprazolam solution was added into vials filled with 5 mL of each sample. Alprazolam concentration in the beverage samples was 400 mg mL^−1^. For electrochemical alprazolam measurements, each beverage sample containing alprazolam was added to the BR buffer to final concentrations of 4, 8, and 16 mg mL^−1^. The standard addition method was also employed to measure alprazolam in three beverage samples. The statistical software SPSS 21 was employed to compare the developed sensor and GC–MS data using the paired t-test. The *p*-value was considered to have no significant difference at the value of >0.05, at the confidence limit of 95%.

## 4. Conclusions

In this paper, an electrochemically pretreated glassy carbon electrode for alprazolam measurement was successfully prepared using a simple electrochemical technique. The combination of the electrochemical pretreatment and the use of AdCSV improved the sensor performance for the detection of alprazolam. The electrochemical pretreatment was especially effective, roughening the surface of the electrode and activating it with the oxygen-containing functional groups of graphite oxide. The pretreated electrode adsorbed more alprazolam than a glassy carbon electrode in the preconcentration step of AdCSV. Determining alprazolam under optimal conditions, the fabricated sensor output two linear ranges (0.1 to 4 and 4 to 20 mg L^−1^) and exhibited high sensitivity, low limits of detection (0.03 mg L^−1^) and quantification (0.1 mg L^−1^), good repeatability, and good recovery. The sensor was used to determine alprazolam in widely available beverages, and the results were in good agreement with results obtained from gas chromatography-mass spectrometry.

## Figures and Tables

**Figure 1 molecules-26-02958-f001:**
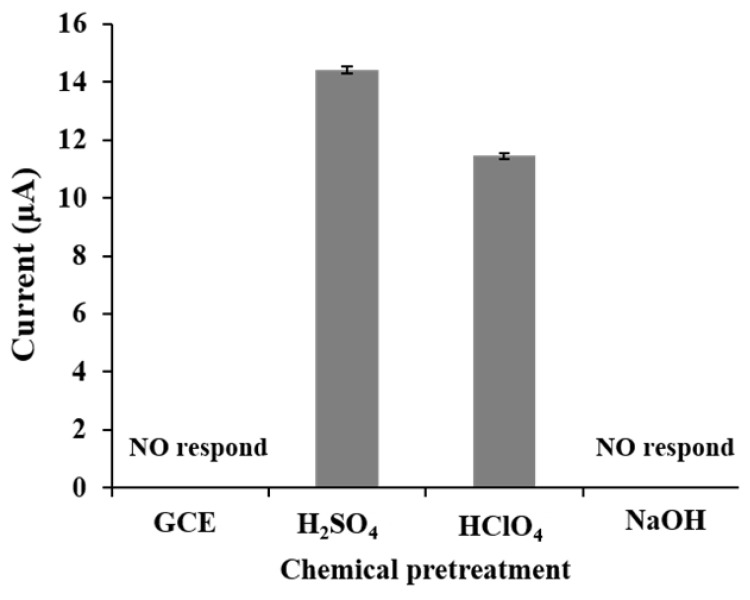
The effect of different pretreatment chemicals on the current response (4 mg L^−1^ alprazolam) of the GCE.

**Figure 2 molecules-26-02958-f002:**
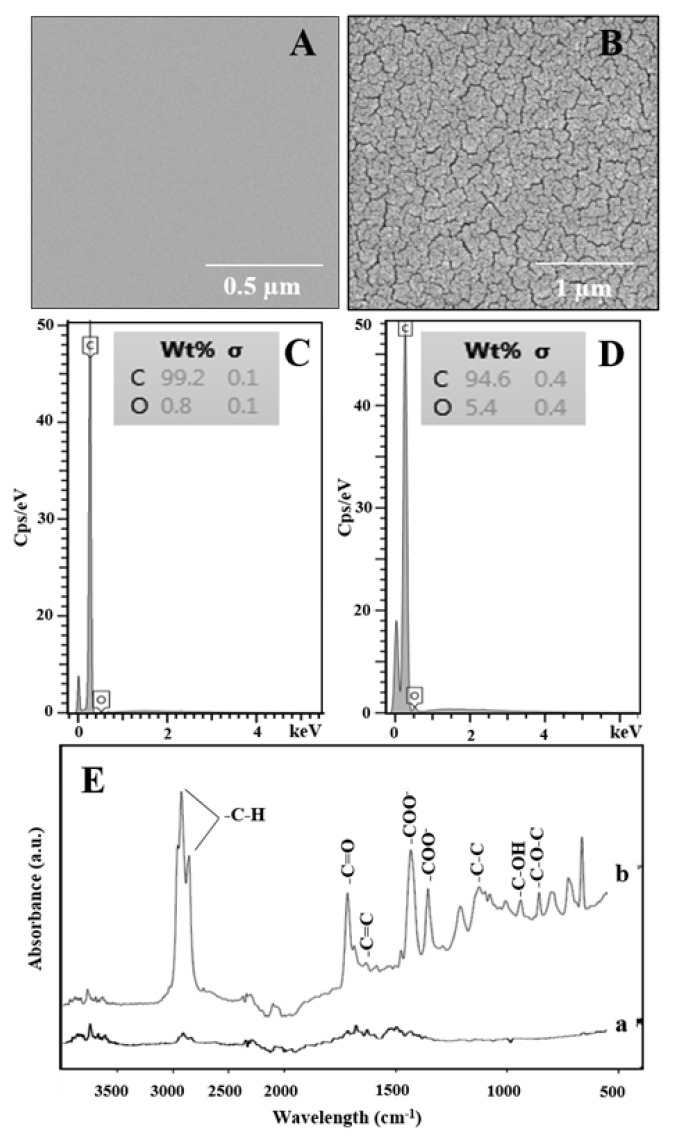
SEM image and EDX spectrum for the GCE (**A**,**C**) and the EPGCE (**B**,**D**). (**E**) FTIR spectra of the GCE (**a**) and the EPGCE (**b**).

**Figure 3 molecules-26-02958-f003:**
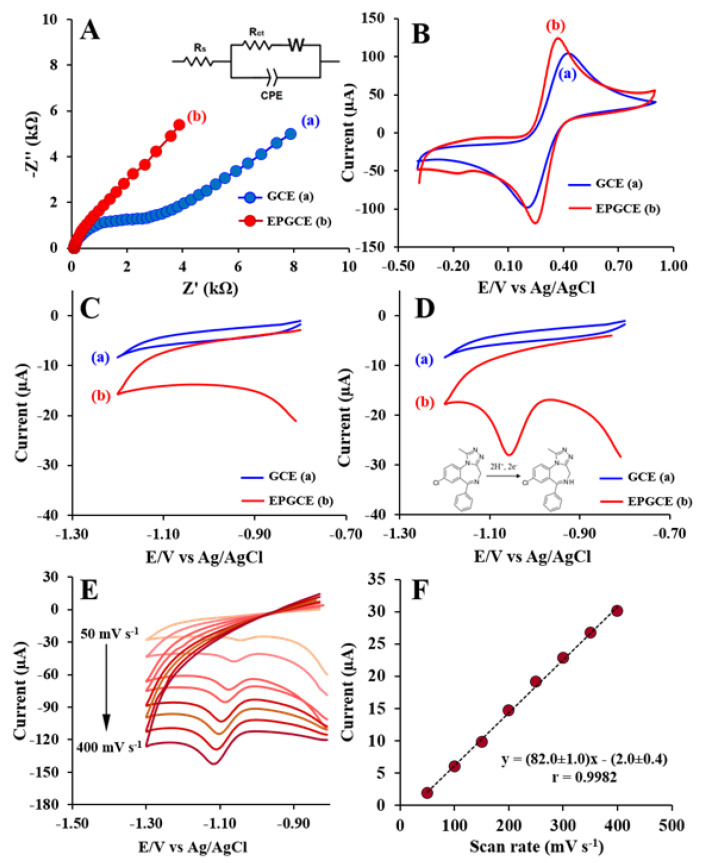
EIS spectra (**A**) and CVs (**B**) of the GCE and the EPGCE in 0.10 M KCl containing 5.0 mM Fe(CN_6_)^3−/4−^ at scan rate 100 mV s^−1^. CVs of the GCE and the EPGCE at a scan rate of 100 mV s^−1^ in BR buffer at pH 9 without alprazolam (**C**) and with 4 mg L^−1^ alprazolam (**D**). CVs from the reduction of 4 mg L^−1^ alprazolam at different scan rates (50–400 mV s^−1^) (**E**). The calibration plot of cathodic current versus scan rate for the EPGCE in BR buffer containing 4 mg L^−1^ alprazolam (**F**).

**Figure 4 molecules-26-02958-f004:**
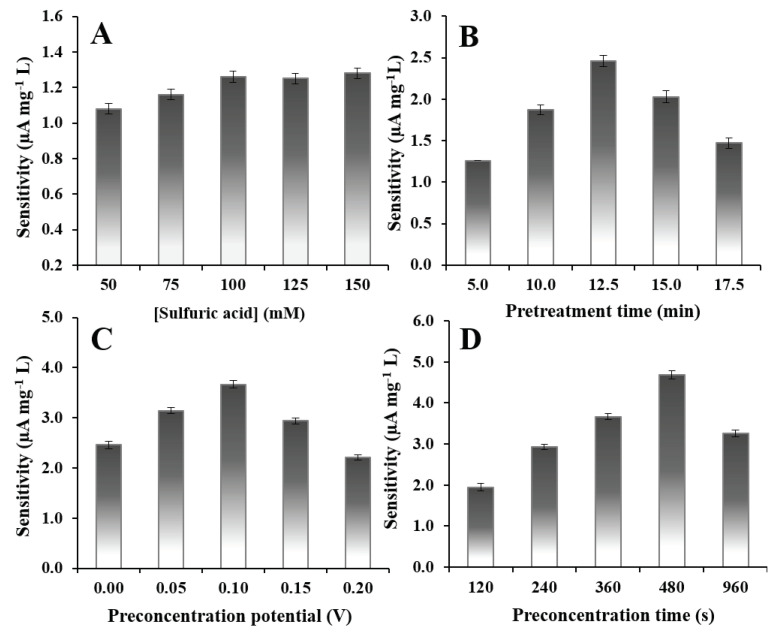
The sensitivity (0.25–4 mg L^−1^ alprazolam) of the sensor (**A**) electrochemically pretreated for 5.00 min at different concentrations of sulfuric acid (preconcentration potential, 0.00 V; preconcentration time, 240 s), (**B**) electrochemically pretreated for different times in 100 mmol L^−1^ sulfuric acid (preconcentration potential, 0.00 V; preconcentration time, 240 s): sensitivity of the sensor at different (**C**) preconcentration potentials for a preconcentration time of 240 s (sensor fabricated in 100 mmol L^−1^ sulfuric acid for a pretreatment duration of 12.50 min) and (**D**) for different preconcentration times at a preconcentration potential of 0.10 V (sensor fabricated in 100 mmol L^−1^ sulfuric acid for a pretreatment time of 12.50 min).

**Figure 5 molecules-26-02958-f005:**
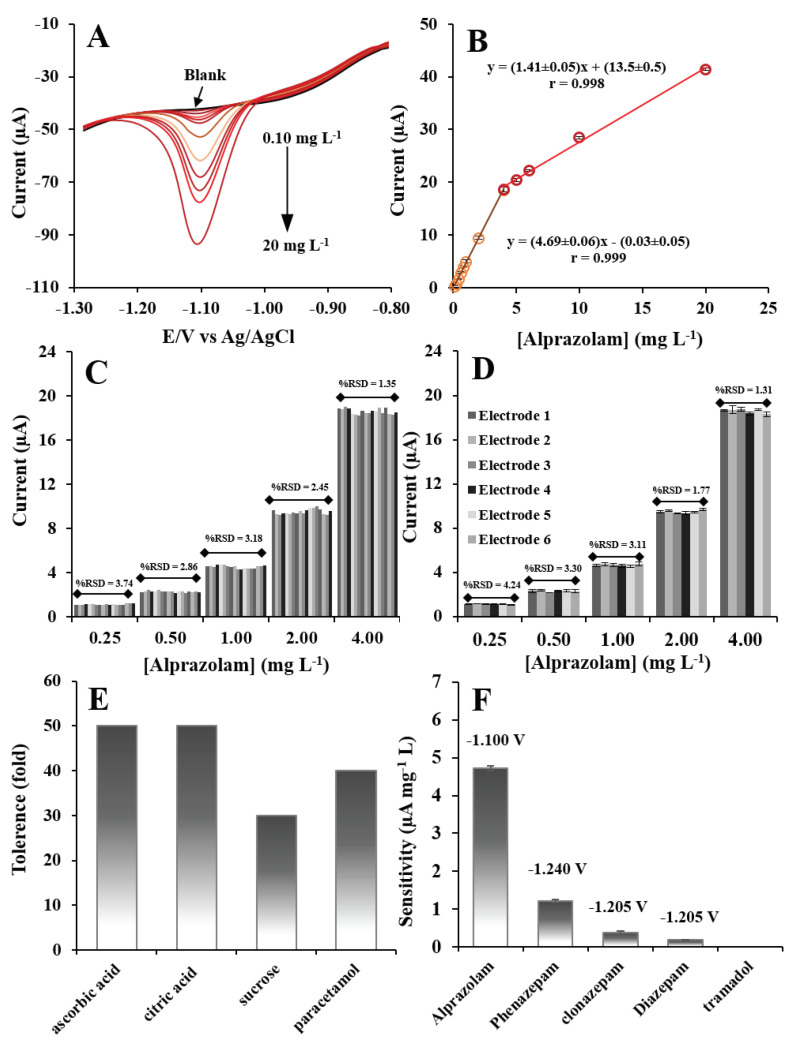
(**A**) AdCSVs at different concentrations of alprazolam. (**B**) The calibration plot between cathodic peak current and alprazolam concentration. The repeatability of the EPGCE preparation for (**C**) the same electrode and (**D**) six electrodes. (**E**) Tolerance limit of interfering compounds (ascorbic acid, citric acid, sucrose, and paracetamol) in alprazolam determination. (**F**) The sensitivity (0.25–4 mg L^−1^ alprazolam) of the developed alprazolam sensor toward interfering species (phenazepam, clonazepam, diazepam, and tramadol).

**Figure 6 molecules-26-02958-f006:**
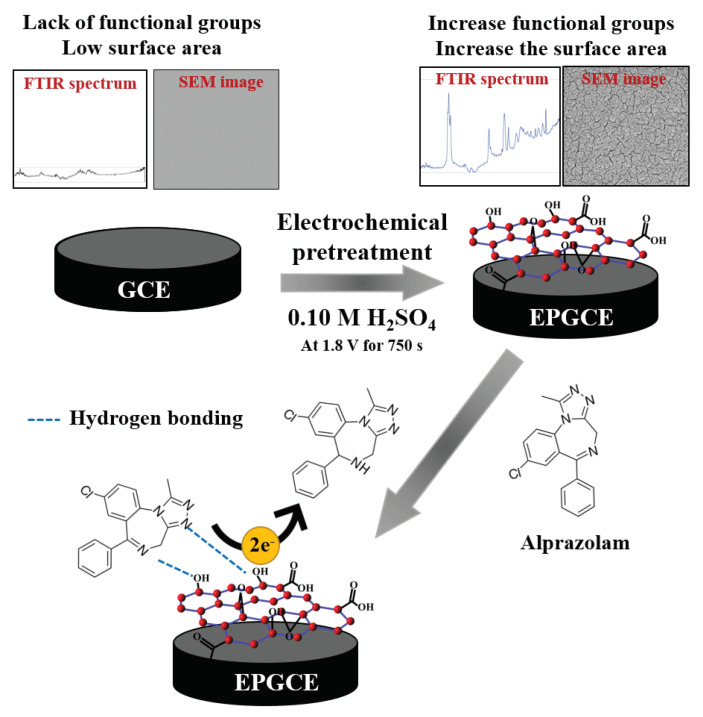
Schematic representation of the preparation of the EPGCE.

**Table 1 molecules-26-02958-t001:** Comparison of analytical performances of the developed sensor and other techniques for the measurement of alprazolam.

Modified Electrode	Technique	Linear Range (mg L^−1^)	LOD (mg L^−1^)	Sample	Reference
^a^ CPE	Potentiometry	0.310–3087.70	0.300	Pharmaceutical tablets	[1]
^b^ BDDE	^f^ DPV	0.250–30.9	0.198	Pharmaceutical tablets	[11]
^c^ m-AgSAE	DPV	0.185–30.9	0.155	Urine	[37]
CPE	DPV	0.247–30.9	0.130	Pharmaceutical tablets	[10]
^d^ EPGCE	^g^ AdCSV	0.100–20.0	0.03	Pepsi, coke, orange Juice, beer, wine, vodka	This work
Other method					
^e^ GC–MS	-	50–1000	7.00	Beer and peach juice	[42]
UV visible spectrometry	-	1.00–20.0	0.400	Pharmaceutical tablets	[43]

^a^ CPE: carbon paste electrode. ^b^ BDDE: boron-doped diamond electrode. ^c^ m-AgSAE: meniscus-modified silver solid amalgam electrode. ^d^ EPGCE: electrochemically pretreated glassy carbon electrode. ^e^ GC–MS: gas chromatography-mass spectrometry. ^f^ DPV: differential pulse voltammetry. ^g^ AdCSV: adsorptive cathodic stripping voltammetry.

**Table 2 molecules-26-02958-t002:** The recovery analysis of alprazolam in beverage samples.

Sample	%Recovery of Proposed Method (n = 3)		
	Concentration of Spiking (mg L^−1^)		
	4	8	16
Pepsi Max Taste	97.0 ± 0.2	99.1 ± 0.3	104.3 ± 0.7
Smirnoff Black Ice	91.8 ± 0.3	100.2 ± 0.1	106.8 ± 0.4
Eristoff vodka	82.0 ± 0.2	107.5 ± 0.3	101.8 ± 0.3
Coke Light	86.90 ± 0.04	95.4 ± 0.4	109.0 ± 0.3
Orange Big	98.1 ± 0.5	102.9 ± 0.4	98.8 ± 1.2
Full Moon Wine	87.4 ± 0.2	93.5 ± 0.1	106.8 ± 0.5

**Table 3 molecules-26-02958-t003:** Comparison of the results of the proposed method and GC–MS method for determination of alprazolam in beverage samples.

Sample	Alprazolam Spike (mg L^−1^)	GC–MS Method Found (mg L^−1^) (n = 3)	Proposed Method Found (mg L^−1^) (n = 3)
Pepsi Max Taste	4	4.4 ± 0.2	3.46 ± 0.15
Eristoff vodka	16	17.3 ± 0.8	17.35 ± 0.17
Smirnoff Black Ice	40	36.9 ± 3.5	39.84 ± 0.88

## Data Availability

The data presented in this study are available on request from the corresponding author.

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
