# Peer review of "Adsorptive Cathodic Stripping Voltammetry for Quantification of Alprazolam"

_molecules, 2021, doi:10.3390/molecules26102958_

Round 1

Reviewer 1 Report

In this work the authors activate the surface of glassy carbon (GC) electrodes by electrochemical oxidation and use the obtained material for the detection of alprazolam (a benzodiazepin) by adsorptive cathodic stripping voltammetry (CSV). Electrochemical activation of  GC (indicated as EPGCE by the authors)  and CSV detection are known and applied for the detection of a variety of organic molecules of biological and pharmaceutical interest. Here the authors combine for the first time these techniques to detect alprazolam, however the research does not seem particularly new since it is the combination of techniques already known, here applied for a very specific analytical purpose. This makes the paper suited for eventual publication on a specialized analytical journal but not on a journal dedicated to chemical topics of wide interest and broad perspectives such as Molecules.  

Moreover, as explained in detail below, there are some points in the paper which need careful revision and further discussion.  In conclusion, my opinion is that, after revision, the paper can be resubmitted for publication on a more specialized journal, such as Analytica or Chemosensors.

Specific comments

  • Lines 57-58. The references cited and discussed in the introductory presentation of the electrochemical activation of GC is quite limited and some important references are missing such as the fundamental review by R. McCreery, Chem Rev. 220, 108, 2646-2687 and a couple of recent papers, see Catalysis today, 2017, 295, 32-40; ChemElectroChem 2015, 2, 761-767, strictly related to the present work.
  • Lines 84-87. The above lacks on the state of the art, are reflected in the discussion concerning the effects of the oxidation of GC in sulfuric acid at 1.8 V. Supporting the formation of functional groups containing oxygen only on the basis of EDX measurements (not very sensitive and not suitable for quantitative measurements) is insufficient and arguable. The same issue was indeed studied in detail in the above uncited papers by voltammetry, Raman, IR, XPS, EIS, providing set of data useful for the precise and quantitative evaluation of the functional groups produced by the EC oxidation of GC.
  • Figures 1 D and F. As expected, the oxidation of GC causes a dramatic increase of the electrode area (roughly 3 times larger than untreated GC) so that both the faradic and capacitive currents increase. Note that the capacitive current in voltammetry, is the main source of background noise, therefore a high capacitive current is not usually welcome for analytical purposes. First of all, I think that the authors should comment on this. Secondly, I wonder why the authors didn’t   use differential pulse voltammetry (DPV) for further improving the detection, since DPV is usually used to lower the noise caused by a high capacitive current.
  • Section 2.6. Analytical performances. The authors use activated GC, but it is not written if the activation is performed before each measurement, that is before obtaining each point of the calibration plot in Fig. 3 B.
  • In Figure 3D, it would be useful to report the standard deviation also for 6 repeated activations of the GC at zero analyte concentration (blank signals). Since the activation is crucial in the present method, in my opinion, this is the blank starndard deviaton to be used to calculate the detection limit, rather than the standard deviation of the intercept.
  • Interference by other benzodiazepins. In Fig. 3F, the authors compare the sensitivity for alprazolam with other similar drugs and on line 223-224 they conclude that there is no interference. This looks too optimistic. For the case of phenazepam, even assuming that the signal of alprazolam (at 1.1 V) is separated enough from that of phenazepam (at -1.24 V), from the sensitivities it is clear that a 3-fold excess of phenazepam will cause the lowering of the alprazolam peak of 50%, since ther will be competition for adsorption sites. This does not look to be a negligible interference effect. Moreover, to confirm that the peak of the main analyte is well separated from the one of the interference, the authors should show and compare the voltammograms for alprazolam in the absence and in the presence of equimolar and excess phenazepam.
  • Real samples-page 9. Is the quantification by CSV performed using external calibration plot or by standard addition method? Again, is it necessary to perform the activation of GC before each measurement or standard addition?
  • Section 3.3. Different types of GC are commercialized by different companies. For instance Tokai offers GC 20, 30 etc. Please, specify the provider and kind of GC used here.

Minor points.

  1. Please, define all abbrevaitions the first time you use them. Unify   abbreviations (e.g. CSV and AdCSV).
  2. Some figures are too small and difficult to read.

Author Response

Please find attached the point-by-point responses to reviewer#1 comments.

Reviewer 2 Report

The manuscript describes the use of a pre-treated GC electrode for the quantification of alprazolam by AdCSV. However, the manuscript needs to undergo some modifications before being accepted.

1) p. 2 lines 63-64

The authors describe that functional oxygen groups on the GC surface can easily interact with alprazolam via π - π interaction and hydrogen bonds. 

I believe that this information is not correct. The presence of functional oxygen groups will hinder Pi-Pi interactions. An analogy would be to obtain graphene oxide (soluble in water) and graphene (insoluble in water).
Thus, the authors must rethink the justification they gave throughout the manuscript.

2) Figure 1

The CVs for the pretreated electrode in the presence of the analyte are incomplete. I would like to verify the complete electrochemical response of the process.

In addition, the comparison between the electrodes is not adequate.

Author Response

Please find attached the point-by-point responses to reviewer#2 comments.

Reviewer 3 Report

Recently, analytical chemistry has focused on improvements in sensitivity, selectivity, and accuracy or enhancement to the new field of application. Electrochemical sensors have recently found their valuable applications in different analytical purposes in food, pharmaceutical, environmental, or clinical analysis. Hence, a new analytical method for the determination of alprazolam seems to be interesting and has a significant clinical interest. The instrumentation and methodology used by the authors were well-chosen, properly designed and conducted but the instrumentation and methodology used by the authors are neither new or innovative. All sections are logical, the discussion is based on recent literature and the results presented in this paper are well prepared. However, there are still some significant aspects that authors should improve.

  1. First of all, the modification of glassy carbon electrode has been previously, extensively described in the available literature including electrochemical pretreatment. Moreover, it seems that clear improvements in sensitivity have not been reached. The authors intentionally (or not) confuse the LOD with LOQ in some cases comparing the analytical performances of the proposed system with performances of other suggested techniques for the determination of alprazolam. The method for the estimation of limits (LOD and LOQ) should be carefully described, it seems that the use of modern well-recognized guidelines (i.e. ICH) should be used instead of the old IUPAC definition. 
  2. Interferences. Why the selectivity has been checked not including the influence of methanol, ethanol or different sweeteners present in the investigated samples? Real samples are likely spiked samples.
  3. Normally, I have expected DPV or SQWV technique for the quantification of analyte it seems that in this work LSV has been applied for this purpose without justification. Hence, the AdSV procedure has not been sufficiently described, the authors should provide more information on the important parameters.
  4. In my opinion, there are many aspects that authors should improve such as the effect of pH, the surface area of the electrode and the type of supporting electrolyte on the alprazolam peaks. 
  5. The comparison with the GC-MS reference method should be described in detail with the statistical results and the interpretation.
  6. All figures presented (particularly Figures 1 A – C) are blurred and it is hard to read them. 
  7. Line 47: Please use proper abbreviation - should be AdSV (ASV - Anodic Stripping Voltammetry)

In my opinion, the manuscript is unsuitable for publication in its current form and requires major revision before any other consideration.

Suggested literature:

Nunes, C.N.; Pauluk, L.E.; Dos Anjos, V.E.; Lopes, M.C.; Quináia, S.P. New approach to the determination of contaminants of emerging concern in natural water: study of alprazolam employing adsorptive cathodic stripping voltammetry. Anal. Bioanal. Chem. 2015, 407, 6171–6179, doi:10.1007/s00216-015-8792-1.

Honeychurch, K.C. Review of electroanalytical-based approaches for the determination of benzodiazepines. Biosensors 2019, 9.

Author Response

Please find attached the point-by-point responses to reviewer#3 comments.

Round 2

Reviewer 2 Report

The authors made all the corrections suggested in the manuscript.

Reviewer 3 Report

Dear Authors,

Thank you for the incorporation of my suggestions. There are some aspects that could be corrected but without them, the manuscript does not lose its attractiveness and sound scientific good.